# A Pervasive History of Gene Flow in Madagascar’s True Lemurs (Genus *Eulemur*)

**DOI:** 10.3390/genes14061130

**Published:** 2023-05-23

**Authors:** Kathryn M. Everson, Mariah E. Donohue, David W. Weisrock

**Affiliations:** 1Department of Integrative Biology, Oregon State University, Corvallis, OR 97331, USA; 2Department of Biology, University of Kentucky, Lexington, KY 40506, USA; mariah.donohue@uky.edu (M.E.D.); david.weisrock@uky.edu (D.W.W.)

**Keywords:** gene flow, *Eulemur*, Madagascar, phylogenetics, primates, systematics

## Abstract

In recent years, it has become widely accepted that interspecific gene flow is common across the Tree of Life. Questions remain about how species boundaries can be maintained in the face of high levels of gene flow and how phylogeneticists should account for reticulation in their analyses. The true lemurs of Madagascar (genus *Eulemur*, 12 species) provide a unique opportunity to explore these questions, as they form a recent radiation with at least five active hybrid zones. Here, we present new analyses of a mitochondrial dataset with hundreds of individuals in the genus *Eulemur*, as well as a nuclear dataset containing hundreds of genetic loci for a small number of individuals. Traditional coalescent-based phylogenetic analyses of both datasets reveal that not all recognized species are monophyletic. Using network-based approaches, we also find that a species tree containing between one and three ancient reticulations is supported by strong evidence. Together, these results suggest that hybridization has been a prominent feature of the genus *Eulemur* in both the past and present. We also recommend that greater taxonomic attention should be paid to this group so that geographic boundaries and conservation priorities can be better established.

## 1. Introduction

The field of systematics is undergoing a paradigm shift regarding the relationship between species boundaries and reproductive isolation. As more studies have identified hybridization and gene flow across the Tree of Life [1,2,3,4], it has become more widely appreciated that “good” species (which might be defined by a wide variety of species concepts [5]) can exist even where reproductive isolation is gradual, impermanent, or porous [6]. Indeed, while hybridization was originally assumed to be rare and to have primarily negative or homogenizing ramifications for species [7,8], it is now recognized as a force that can fuel speciation through reinforcement or the conception of novel allele combinations for selection to act upon [8,9,10,11]. In light of this accumulating knowledge, researchers in the field of phylogenetics are developing new methods for accommodating and parameterizing introgression and/or gene flow [12,13,14,15,16]. These approaches have been shown to produce more accurate estimates of topology, divergence time, and population size in empirical systems [17,18,19].

While hybridization is common throughout the Tree of Life, its study is of particular interest in primates. Genomic evidence continues to reveal that introgressive hybridization among archaic *Homo* species contributed to the evolution of anatomically modern humans by introducing genes associated with brain size, immune function, and other important traits [20,21,22,23,24]. Across the broader primate tree, hybridization seems to be a common present-day phenomenon, as active hybrid zones have been documented in taxa as diverse as howler monkeys [25], baboons [26,27], marmosets [28], gorillas [29], and lemurs [30,31]. Hybridization is also increasingly being seen as a mechanism that could drive the generation of new primate species diversity. For example, ancient hybrid speciation has been hypothesized as the mechanism that gave rise to one species of marmoset [32] and several species of macaque [33,34] and langur [35,36].

For several reasons, the true lemurs of Madagascar (genus *Eulemur*, 12 species) form an excellent primate system that can be used to study the interplay between species limits and gene flow: (a) *Eulemur* experienced recent, rapid radiation, with a known history of hybridization [37,38]; (b) excellent resources are available for their genetics, geographic range, diet, and social behavior [39,40,41,42]; (c) the study of their biodiversity is important in light of urgent conservation concerns [43,44]. For many years, *Eulemur* was assumed to be a species-limited clade and was dismissed by many as Madagascar’s “little brown lemurs” (a nod to “little brown birds” à la ornithology) [45,46]. However, over time, greater systematic attention has uncovered a higher level of diversity within this clade than that which was previously realized, with 12 species recognized today (an increase from only five species recognized in the year 2000). Researchers have also discovered at least five known active hybrid zones in the genus: *E. mongoz x E. rufus* in western Madagascar [47], *E. flavifrons x E. macaco* in the north [48,49], *E. albifrons x E. sanfordi* in the north-east [48], *E. rufus x E. collaris* in the south-west [50,51]. and *E. rufifrons x E. cinereiceps* in the south-east [30,31,52,53,54].

The largest studies of *Eulemur* to date in terms of taxonomic sampling have been conducted by Markolf et al. [38,55], who used a dataset containing all 12 species and four genetic loci. These studies found that there is substantial admixture among species, as well as uncertainty in the deepest nodes of the *Eulemur* phylogeny; however, all species had at least one line of evidence, indicating that they were valid species. We note that “valid species”, in this case, were defined using a variety of data types (genetic ancestry, unique morphological traits, or unique calls), but all could be identified as independently evolving metapopulations under the unified species concept [56]. In this study, we readdress the evolutionary history of *Eulemur* using a large, multilocus genetic dataset, and we leverage new phylogenetic methods that explicitly model trees as reticulating networks. These results cast a new light on the genus by revealing that hybridization and gene flow within *Eulemur* is not merely a recent phenomenon among extant species, but it has been a pervasive feature throughout their evolutionary history.

## 2. Materials and Methods

Nuclear DNA sequence data from 13 individuals (all 12 species of *Eulemur* plus 1 outgroup species, *Lemur catta*) were downloaded from the National Center for Biotechnology Information (NCBI) Sequence Read Archive (BioProject ID PRJNA957840; Appendix A). Full details on how these data were generated can be found in the original study by Everson et al. 2023 [57]. Briefly, the samples were obtained as frozen blood or tissue samples (Appendix A), and genomic DNA was extracted using the Qiagen DNEasy Blood and Tissue Kit (Qiagen, Inc., Hilden, Germany). Library preparation and sequencing were performed in the Translational Lab in the College of Medicine at Florida State University following the Anchored Hybrid Enrichment (AHE) protocol [58], which targets hundreds of single-copy loci across the genome that contain phylogenetic information at both deep and shallow time scales. The probes used for library preparation were adapted from the Amniote 2 probe set [59], but they were modified to produce longer loci and to target primate DNA more closely. Sequencing was performed on an Illumina NovaSeq 6000 instrument using paired-end 150 base-pair chemistry. Reads were demultiplexed and quality filtered using the Illumina CASAVA software (Illumina, Inc.), and additional quality control and read merging techniques were performed following the methods outlined in Rokyta et al. 2012 [60].

The same nuclear loci were also extracted from five previously published *Eulemur* genomes available from NCBI: *E. flavifrons* (SAMN03699688), *E. fulvus* (SAMN07678083), *E. macaco* (SAMN03699689), *E. rubriventer* (SAMN09702726), and *E. rufifrons* (SAMN08793327). We used BLAST [http://www.ncbi.nlm.nih.gov/blast (accessed on 19 April 2023)] to map and extract regions matching our probe sequences from each genome, as described in [57]. We aligned all nuclear sequence data (the 5 previously published genome individuals plus the 13 individuals derived from AHE) from each locus using MAFFT v7.023b [61]. Finally, all loci were visualized using Geneious Prime v.2023.0.1 software (Biomatters Ltd., Aukland, New Zealand) for the final quality check. Poorly aligned regions were realigned using the Local Realignment tool, and the final alignments were exported for further analysis. 

We also downloaded previously published mitochondrial D-Loop (control region) sequence data from 479 individuals through NCBI’s GenBank database (Appendix A). These data were again aligned using Geneious Prime. In some cases, the taxonomy of *Eulemur* samples had changed since their deposition on GenBank. We manually updated these cases according to the following recent taxonomic changes: six former subspecies of *E. fulvus* (*E. f. albifrons*, *E. f. cinereiceps*, *E. f. collaris*, *E. f. fulvus*, *E. f. rufus*, and *E. f. sanfordi*) were elevated to full species [62]; *E. macaco flavifrons* was elevated to *E. flavifrons* [63]; *E. rufus* was split into *E. rufus* and *E. rufifrons* [63,64]; *E. albocollaris* became a junior synonym of *E. cinereiceps* [65].

Both the mitochondrial dataset and the concatenated nuclear dataset were used to estimate maximum likelihood phylogenies using the IQ-TREE webserver, which invokes IQ-TREE v.1.6.12 [66]. In both cases, the substitution model was estimated automatically using ModelFinder [67], and node support was assessed using 1000 ultrafast bootstrap replicates [68]. We also estimated a species tree using the nuclear dataset in SVD-Quartets [69], which was performed in PAUP* v.4a168 [70] using the concatenated nuclear sequence file as the input. We evaluated all quartets (evalq = all) and used the multilocus bootstrapping method with 1000 replicates to assess branch support. Finally, we also estimated a species tree from the nuclear dataset using the ASTRAL v.5.7.8 program [71]. In order to run this program, first, we generated individual gene trees from each locus using RAxML-ng v.1.1.0 [72] under the GTR model. These trees were used as the input to ASTRAL, and all other parameters were set to default values. Across all of the above analyses, *Lemur catta* was designated as the outgroup.

To visualize our nuclear data as a network, we first used SplitsTree v.4.14.8 [73], which generates a splits graph using genetic distances. We ran this analysis with uncorrected p-distances and the NeighborNet algorithm [74]. For more explicit coalescent-based phylogenetic network analysis, we then used the SNaQ model within the julia package PhyloNetworks [75]. This method uses a summary multi-species coalescent model to simultaneously estimate the species tree, while also allowing reticulate or hybrid edges. Using the maximum pseudolikelihood approach within SNaQ [76], we used the gene trees previously estimated with RAxML-ng as the input and estimated six models that varied in the maximum number of hybrid edges (hmax) from hmax = 0 to hmax = 5. The optimal number of hybrid edges was selected by visualizing the change in negative log pseudolikelihood (−logplik) scores across all models. 

## 3. Results

### 3.1. Mitochondrial Phylogeny

The final mitochondrial dataset containing 914 aligned base pairs (bp) was used to generate a maximum likelihood phylogeny that was reasonably well-resolved, although several described species were not recovered as monophyletic groups (Figure 1, Appendix A). Specifically, we found that *E. albifrons*, *E. sanfordi*, and *E. fulvus* formed a single clade, without obvious species boundaries. Likewise, we found that *E. rufus* and *E. rufifrons* were not reciprocally monophyletic. Finally, while we did identify a clade representing *E. cinereiceps,* we note that 10 individuals of *E. cinereiceps* fell within the *E. rufus* + *E. rufifrons* complex (this may not be particularly surprising, given that *E. cinereiceps* and *E. rufifrons* are known to hybridize in the wild [30,31,52,53,54]).

### 3.2. Nuclear Species Tree

Our final nuclear dataset contained 331 loci with an average per-locus length of 3339 bp and a total concatenated alignment length of 1,108,850 bp. The level of bootstrap support on the concatenated nuclear maximum likelihood tree was high overall (all nodes ≥ 95%; Figure 2a), although once again, *E. rufifrons* and *E. fulvus* were paraphyletic. We could not make inferences about the monophyly of *E. albifrons*, *E. sanfordi*, and *E. rufus*, as our nuclear dataset contained only one sample from each species.

Our two species tree analyses (estimated using ASTRAL and SVD-Quartets) recovered non-identical topologies (Figure 2b,c). The ASTRAL topology recovered *E. rubriventer* as a sister to all other *Eulemur*, whereas the SVD-Quartets topology recovered *E. mongoz* as a sister to all other *Eulemur*. All other relationships were identical. Most nodes on both trees received maximum support (quadripartition support values of 1.0 or Bootstrap support values of 100 in the ASTRAL and SVD-Quartets trees, respectively). However, the node splitting E. coronatus from E. flavifrons and E. macaco received less support in both trees (0.82 and 74), and ASTRAL recovered an extremely short internode branch length from the prior node (note that branch lengths were not estimated in the SVD-Quartets analysis). Some relationships within the *E. fulvus* + *E. rufifrons* + *E. rufus* + *E. collaris* + *E. cinereiceps* clade also received less support (ASTRAL: 0.56–1.0 quadripartition support and SVD-Quartets: 88–100 bootstrap support in this clade) and included several short branches in the ASTRAL analysis.

### 3.3. Phylogenetic Networks

The network estimated using SplitsTree included distinct branches or sets of branches for each species, with the exception of *E. fulvus*, which did not form a single monophyletic species group (Figure 3). Moreover, this network exhibited several reticulations, suggesting that gene flow may be present in the genus. Shallow, reticulated branches were particularly evident in the complex containing *E. fulvus, E. rufifrons, E. rufus, E. cinereiceps,* and *E. collaris*, and in the center of the network. In our SNaQ analysis, the top-supported model (that with the lowest log-likelihood value) had three hybrid edges (*H* = 3; Figure 4a). However, we note there was minimal improvement after one hybrid edge; so, we consider models with one or two hybrid edges to also be well supported. The model with one hybrid edge estimated a reticulation between *E. mongoz* and the clade containing *E. flavifrons* and *E. macaco*. The model with two hybrid edges added a reticulation between *E. rufus* and *E. cinereiceps*. Finally, the model with three hybrid edges also included a reticulation between *E. albifrons* and the base of the *E. rufus* + *E. rufifrons* + *E. cinereiceps* + *E. collaris* clade. In the model with three hybrid edges, the minor inheritance probabilities (γ; the proportion of genetic data inherited from the second parental lineage) were γ = 0.17 for the hybrid branch associated with *E. mongoz*, γ = 0.22 for the hybrid branch associated with *E. albifrons*, and γ = 0.49 for the hybrid branch associated with *E. cinereiceps*. Thus, in these instances, a substantial proportion of each genome (between 17 and 49%) may be of hybrid origin.

Importantly, SNaQ was also used to estimate an underlying species tree topology that was non-identical to any of the other topologies estimated in previous analyses. Here, *E. rubriventer* was sister to all other *Eulemur* (similar to the mitochondrial phylogeny and the ASTRAL species tree), and *E. mongoz* was sister to the *E. fulvus + E. rufus* + *E. rufifrons* + *E. cinereiceps* + *E. collaris* clade. The latter relationship was seen in our mitochondrial phylogeny, but was not recovered by those conducting other nuclear analyses.

## 4. Discussion

Until 1998, the scientific consensus was that the genus *Eulemur* contained five species. This number has increased dramatically in recent years, with 12 species recognized today, which exhibit a remarkable display of morphological, behavioral, and ecological variation. Despite these taxonomic advances, resolving the phylogenetic relationships among *Eulemur* species has continued to pose challenges. It was not clear in previous work whether the lack of phylogenetic resolution was due to the use of poorly informative genetic loci, or if it was a reflection of real heterogeneity across the genome (i.e., conflict caused by gene flow or incomplete lineage sorting). In this study, we used a massive, multilocus dataset containing genetic information from many regions of the genome; yet, we were still unable to confidently resolve some regions of the *Eulemur* tree. Specifically, our two species tree methods estimated different topologies (Figure 2), and both of these topologies differed from the mitochondrial phylogeny (Figure 1) and from the topology estimated via SNaQ (Figure 4).

Our analyses revealed one important reason why phylogenetic analyses have failed: gene flow has been a prevalent feature in the genus *Eulemur* throughout its evolutionary history. Most coalescent-based phylogenetic approaches assume that gene tree discordance results entirely from incomplete lineage sorting, but this assumption is violated in systems with a known history of gene flow. While there are at least five examples of active hybrid zones within *Eulemur* [30,31,47,48,49,50,51,52,53,54], our study revealed that introgression is not merely a recent phenomenon, but it has also been present at deeper time scales. Our SNaQ analysis (Figure 4b), as well as the starburst-like pattern at the center of our SplitsTree network (Figure 3), suggest that an ancient introgression event involving *Eulemur mongoz* might be responsible for the difficulty in placing this taxon. At the same time, more recent introgression has likely been the source of poor resolution within the *E. rufus* + *E. rufifrons* + *E. cinereiceps* + *E. collaris* clade (Figure 4b). In light of these findings, we feel that phylogenetic network analyses likely perform a better job of capturing the history of this species complex. Phylogenetic networks are a rapidly developing area of research; so, we are hopeful that the evolutionary history of *Eulemur* and other systems can be reconstructed with finer detail and higher confidence in the coming years as these methods continue to improve. 

Our results also suggest that the genus *Eulemur* is still in need of taxonomic revision, as both mitochondrial and nuclear data point to the non-monophyly of *E. rufus* with respect to *E. rufifrons*. Interestingly, Markolf et al. 2013 [55] noted that these species could be distinguished based on pelage and the loud calls they make, but they could not be distinguished based on other morphological characters. This may simply be a recent divergence event, where an insufficient amount of time has passed for diagnostic differences and genetic mutations to accumulate. Alternatively, it could be that the species limits need to be refined via more detailed geographic sampling. *Eulemur rufifrons* has a disjunct distribution, with some populations in the eastern rainforest and some in the western dry forest, and only the western population is parapatric with *E. rufus*. It is possible that the eastern and western populations of *E. rufifrons,* in fact, represent distinct species, and/or that the western population of *E. rufifrons* should be synonymized with *E. rufus*. These taxonomic decisions are beyond the scope of this study. 

Our mitochondrial dataset also evidenced non-monophyly within the *E. fulvus* + *E. sanfordi* + *E. albifrons* complex. This is an intriguing result, as Markolf et al. 2013 [55] noted that all three of these species can be distinguished on the basis of pelage (although they could not distinguish the species on the basis of acoustic calls). This result also conflicts with our nuclear evidence, i.e., all of our nuclear phylogenies recovered *E. fulvus* as a sister to the *E. rufus* + *E. rufifrons* + *E. cinereiceps* + *E. collaris* clade (but note that *E. fulvus* was not monophyletic in our SplitsTree network (Figure 3)). It is possible that our result of mitochondrial non-monophyly among *E. fulvus*, *E. sanfordi*. and *E. albifrons* could be explained by introgression, which may have led to discordance between the nuclear and mitochondrial genomes of these species. Indeed, we found evidence for gene flow between *E. albifrons* and an ancestral lineage in the *E. fulvus* clade (Figure 4b), and there is a known active hybrid zone between *E. sanfordi* and *E. albifrons* [48]. Any of these situations could have created opportunities for the introgression of the mitochondrial genome (i.e., mitochondrial capture). Mitochondrial capture, wherein the mitochondrial DNA of one species sweeps through another population and supplants the original mitochondrial genome, can be caused by either adaptive (e.g., if the mitochondria of one species confers a fitness advantage) or non-adaptive (e.g., genetic drift) processes [77]. It has been documented in many systems, but it is most often recognized in hybridizing pairs of species; thus, the *E. fulvus* + *E. sanfordi* + *E. albifrons* system might be an interesting and rare case of mitochondrial capture occurring in a hybridizing trio. It would be informative to learn about the origins of this mitochondrial genome in terms of which species’ mitochondrial genomes have been supplanted and when the original capture event occurred. Again, while our dataset did not have a sufficient sample size to dive deeply into these questions, we recommend that future research focuses on population-level sampling within these species with particular focus on proposed species boundaries and regions of sympatry.

Another intriguing target for future study is the *Eulemur rufifrons/cinereiceps* hybrid complex because it represents a recent (or perhaps ongoing) example of how new *Eulemur* species might form in the face of gene flow. Previous studies have suggested that their hybrid zone—a 70 km long region in south-eastern Madagascar—is an independently evolving unit: the population at the center of the zone is in Hardy–Weinberg equilibrium and has private alleles that are not seen in either parental species [31,53], and the hybrids have diets and an ecological niche unlike those of either parental species [54]. The hybrids are also the strictest frugivores of all brown lemurs, with fruits making up 95% of their diet [30,78], compared to 66% of the diets of both parental species [39,78,79]. All of these characteristics suggest that the hybrid population is, in fact, a new species (although it has yet to be formally described as this), with speculative estimates suggesting that hybridization began ~260–1300 generations ago [53]. If this is true, this would be the only known case of active homoploid hybrid speciation in a primate. Homoploid hybrid speciation, the formation of a third distinct species following hybridization between parental species without a change in chromosome number, is most often studied in plants, but it has only recently started to be recognized in vertebrate groups [80,81]. It is worth mentioning that our SNaQ analysis recovered introgression between *E. cinereiceps* and *E. rufus,* rather than *E. rufifrons*. The fact that we did not recover hybridization between *E. cinereiceps* and *E. rufifrons* in this analysis could be explained by either (a) the hybridization having occurred too recently to create a signal of gene tree conflict or (b) taxonomic uncertainty in the *E. rufifrons*/*E. rufus* complex. Regardless of the reason for this result, we reiterate that the *Eulemur rufus/rufifrons/cinereiceps* complex represents an exciting and important target for additional research. 

A final important implication of this study is that it paves the way for future conservation research on these threatened animals. Madagascar is renowned as a biodiversity hotspot, meaning that it has exceptional biodiversity and endemism, but it is also plagued with anthropogenic habitat loss. This is particularly evident in lemurs, with 94% of species considered to be threatened with extinction, and 90% experiencing population decline [82,83]. Our study revealed instances of gene flow within the genus *Eulemur*, but it will be up to conservation practitioners to assess how gene flow might be impacting population health. In some instances, gene flow can help species by introducing new genetic variants. The most commonly applied example of this phenomenon is genetic rescue, whereby the fitness of small populations at risk of extinction can be increased by translocating individuals from other populations [84]. Then, increased fitness is achieved by decreasing the impact of inbreeding depression by creating more fit hybrids (heterosis) and/or by creating novel adaptive combinations of traits. Genetic rescue has been successfully used to improve conservation outcomes in other systems (e.g., the Florida panther [85] and the prairie chicken [86]). By pointing to cases of viable gene flow in *Eulemur*, we have revealed several potential options for genetic rescue in these lemurs; for example, the critically endangered *E. cinereiceps* could tolerate gene flow from *E. rufifrons* and/or *E. rufus* in the event that it is threatened by severe inbreeding depression. On the other hand, some have raised concerns about the use of genetic rescue, as it could lead to harmful effects, including the loss of local adaptive variation or genetic distinctiveness (genetic swamping) or poor outcomes for the transplanted individuals due to the lack of local adaptation (outbreeding depression) [87]. Again, it will be up to conservation practitioners to decide when these tools are appropriate to use. As the true lemurs continue to face extinction, studies such as this one in the fields of systematics and biodiversity science can provide sorely needed information about how many lemur species exist and the variety of factors impacting their genetic diversity. 

## Figures and Tables

**Figure 1 genes-14-01130-f001:**
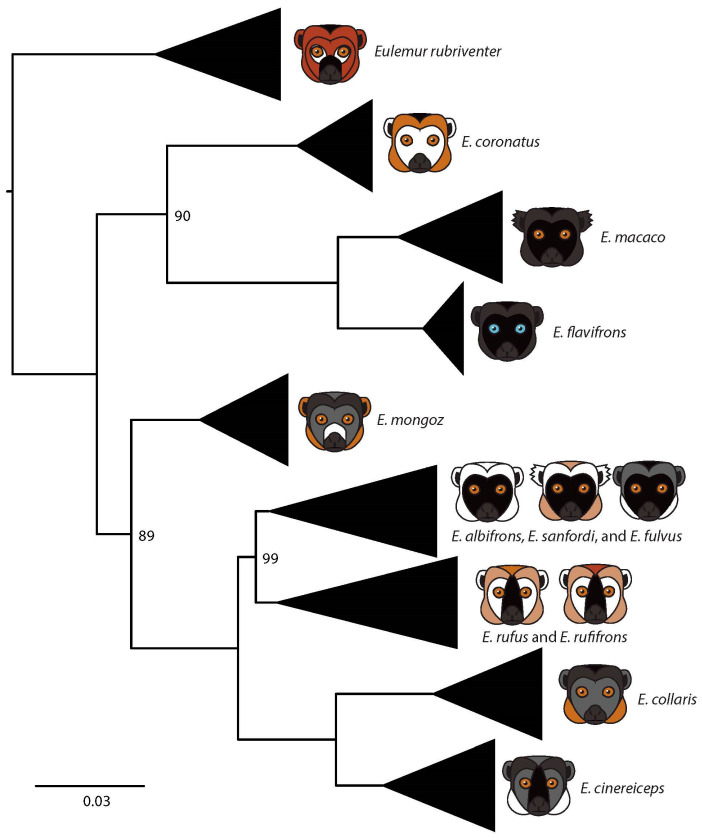
A maximum likelihood phylogenetic tree of *Eulemur* estimated using mitochondrial data. Bootstrap values at each node are 100 unless otherwise indicated. Multiple individuals from each species were included in the analysis, but they are collapsed here for visual clarity (black triangles). The full tree showing all individuals is available in Appendix A. Where multiple species are shown next to a black triangle, those species were not recovered as reciprocally monophyletic. The illustrations associated with each species represent the diagnostic features of males in each species in terms of face, crown, and eye color; presence or absence of a beard; presence or absence of ear tufts. Illustrations were created by K.M.E.

**Figure 2 genes-14-01130-f002:**
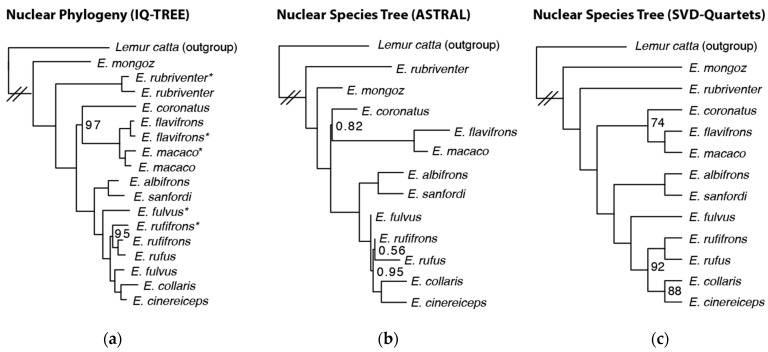
Phylogenetic trees estimated using our nuclear dataset. (**a**) A maximum likelihood phylogenetic tree estimated using IQ-TREE. Bootstrap support values on each node are 100, unless otherwise indicated. Asterisks denote individuals whose nuclear data were pulled from previously published whole genomes (Appendix A). (**b**) A species tree estimated using ASTRAL. Values on nodes are the quadripartition support values and are 1.0 unless otherwise indicated. (**c**) A species tree estimated using SVD-Quartets. Bootstrap support values are 100, unless otherwise indicated. Branch lengths are not scaled to coalescent units.

**Figure 3 genes-14-01130-f003:**
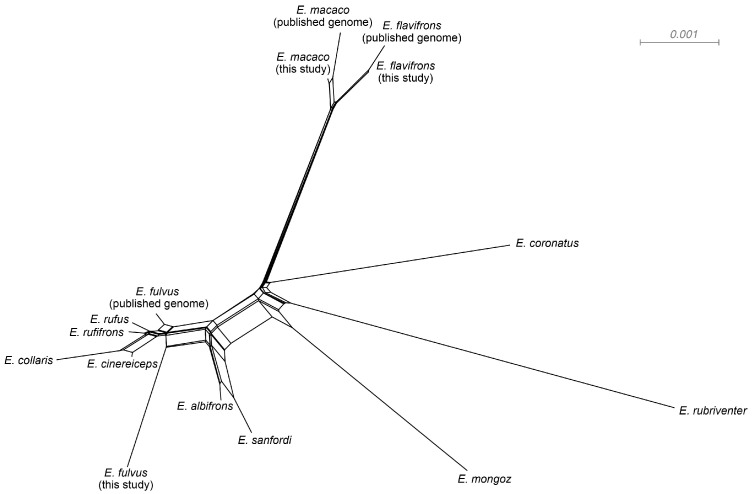
Split decomposition analysis of the nuclear dataset, as estimated using SplitsTree. Visualization of phylogenetic networks was used to identify ambiguities in branching patterns that might be attributed to reticulation.

**Figure 4 genes-14-01130-f004:**
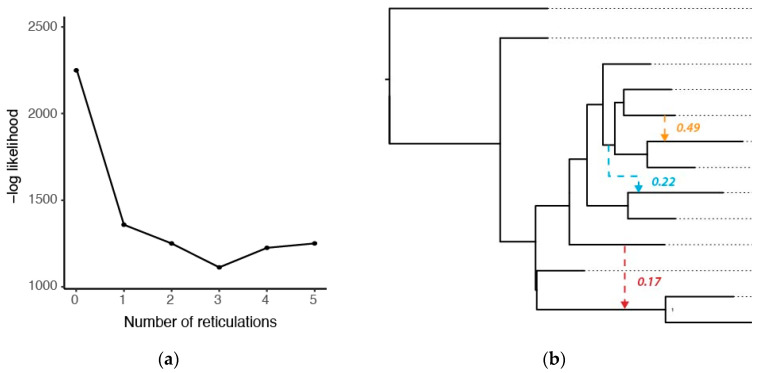
Results from our SNaQ phylogenetic network analysis. (**a**) Negative log-likelihood scores for models with varying numbers of reticulations or hybrid edges (*H* = 0–5). A phylogenetic network with three reticulations was most supported (as indicated by the lowest log-likelihood score), but we explored the results of networks with one, two, and three reticulations. (**b**) The result of the most-supported phylogenetic network (*H* = 3). Dashed lines and colored values denote hybrid edges and their associated minor inheritance probabilities (γ), respectively. The blue hybrid edge was only observed in the *H* = 3 model; the orange hybrid edge was observed in *H* = 2 and *H* = 3 models; the red hybrid edge was observed in models with *H* = 1, 2, or 3.

## Data Availability

Publicly available DNA sequence datasets were analyzed in this study. The NCBI GenBank accession numbers associated with the nuclear and mitochondrial datasets can be found in Appendix A. Computational scripts are openly available on the first author’s github profile: https://github.com/keverson25/EulemurAnalyses.

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
