# Peer review of "A Pervasive History of Gene Flow in Madagascar’s True Lemurs (Genus Eulemur)"

_genes, 2023, doi:10.3390/genes14061130_

Round 1

Reviewer 1 Report

The manuscript present traditional coalescent-based and network-based phylogenetic analyses of nuclear and mitochondrial datasets of the genus Eulemur. The authors found that there were discrepancies between the mitochondrial and nuclear phylogenies, and some species were not recovered as monophyletic. This was particularly evident in the phylogenetic networks. Based on these results, the authors discuss that hybridization has been a prominent feature of the genus at both past and present timescales. The manuscript is well-written, the figures are clear and easy to understand. It was hard to suggest something that could improve the manuscript.

Regarding lines 66-73 in the Materials and Methods section, it may be worth mentioning if there were any constraints by the selection of the nuclear loci used.

Author Response

Reviewer 1: "The manuscript present traditional coalescent-based and network-based phylogenetic analyses of nuclear and mitochondrial datasets of the genus Eulemur. The authors found that there were discrepancies between the mitochondrial and nuclear phylogenies, and some species were not recovered as monophyletic. This was particularly evident in the phylogenetic networks. Based on these results, the authors discuss that hybridization has been a prominent feature of the genus at both past and present timescales. The manuscript is well-written, the figures are clear and easy to understand. It was hard to suggest something that could improve the manuscript."

Response: We thank Reviewer 1 for this kind and positive review.

Reviewer 1: "Regarding lines 66-73 in the Materials and Methods section, it may be worth mentioning if there were any constraints by the selection of the nuclear loci used."

Response: In the new version, we have added approximately 200 words to the Materials and Methods section to provide further information about the properties of our nuclear loci and how these loci were selected.

Reviewer 2 Report

This is an interesting paper. I have no particular concerns, except the following one:

... it has become more widely appreciated that “good” species can exist even where reproductive isolation is gradual, impermanent, or porous.

We have many definitions of species. Could you clarify what you mean by "good species" and which species concept(s) you intend to apply in this study?

Author Response

Reviewer 2: "This is an interesting paper. I have no particular concerns, except the following one:

... it has become more widely appreciated that “good” species can exist even where reproductive isolation is gradual, impermanent, or porous.

We have many definitions of species. Could you clarify what you mean by "good species" and which species concept(s) you intend to apply in this study?"

Response: This is an excellent observation. We added an additional clause and a new citation to clarify what we meant by "good species" in the quoted sentence (paragraph 1 of the introduction). Then, later in the introduction, we also describe the various lines of evidence that have been used for species delimitation in Eulemur in past studies and we qualify that we consider species do be independently evolving metapopulations, as defined by de Queiroz 2007 in the unified species concept. Note that any new attempts to redefine species boundaries in Eulemur are outside the scope of this study, so our own views on species concepts are less relevant to our results and discussion sections.